

# Estimating the impact of lock-down, quarantine and sensitization in a COVID-19 outbreak: lessons from the COVID-19 outbreak in China

Obiora C. Collins[*] and Kevin J. Duffy[*]

Institute of Systems Science, Durban University of Technology, Durban, South Africa
[*] These authors contributed equally to this work.

## ABSTRACT

In recent history, COVID-19 is one of the worst infectious disease outbreaks currently affecting humanity globally. Using real data on the COVID-19 outbreak from 22 January 2020 to 30 March 2020, we developed a mathematical model to investigate the impact of control measures in reducing the spread of the disease. Analyses of the model were carried out to determine the dynamics. The results of the analyses reveal that, using the data from China, implementing all possible control measures best reduced the rate of secondary infections. However, quarantine (isolation) of infectious individuals is shown to have the most dominant effect. This possibility emphasizes the need for extensive testing due to the possible prevalence of asymptomatic COVID-19 cases.

## INTRODUCTION

On 31 December 2019, the World Health Organization (WHO) China Country Office received information of a case of pneumonia detected in Wuhan City, Hubei Province of China (*WHO, 2020c*), and later identified as COVID-19 a disease caused by the virus SARS-CoV-2 (*WHO, 2020b*). The outbreak was associated with exposures to an animal source but it also became clear that the virus spreads between humans.

Globally, COVID-19 resulted in exponential growth in cases and deaths from 22 January 2020 to 30 March 2020 and continued to spread rapidly. Therefore, urgent measures are needed to save humanity from this deadly outbreak.

Figure 1 is a graphical representation of the total confirmed cases of COVID-19 in China from 22 January 2020 to 30 March 2020 (*WHO, 2020d*). Unlike in the global case, the new confirmed cases increased from the onset of the outbreak till 20 February 2020 and then started to decrease. This decrease was due to the implementation of effective control measures, including: public education on disease prevention and environmental hygiene; installation of infra-red thermometers in airports, railway stations, etc. to detect infectious

Corresponding author
Kevin J. Duffy, kevind@dut.ac.za

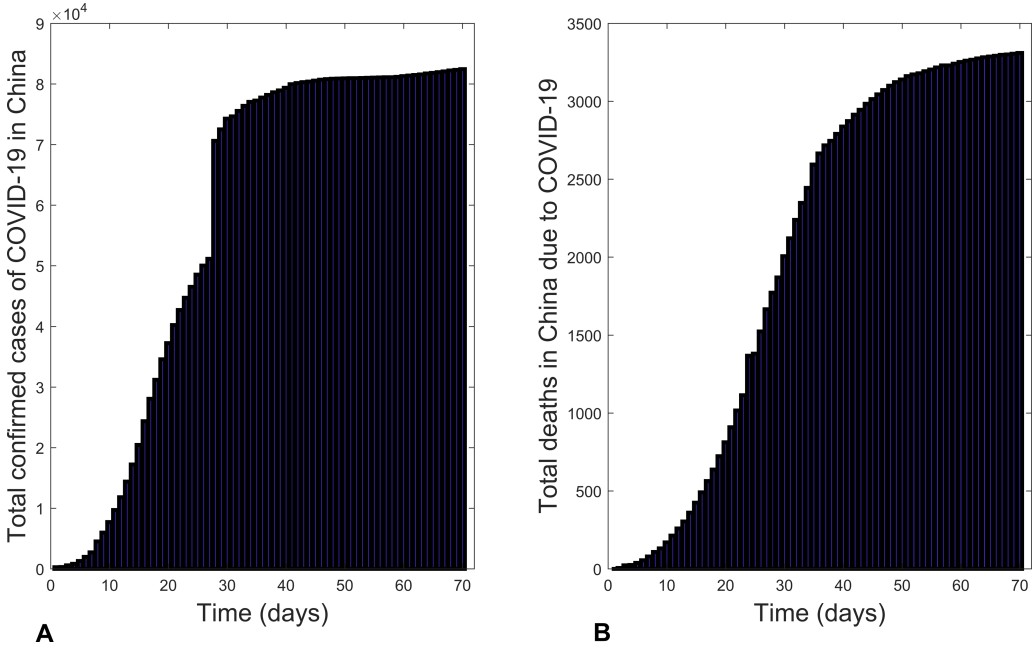

**Figure 1** Bar charts illustrating: (A) the total confirmed cases of COVID-19 in China and (B) total deaths due to COVID-19 in China from 22 January 2020 to 30 March 2020.

individuals for prompt treatment; conducting active case findings in all of China, prompt isolation and treatment of infectious individuals (*WHO, 2020d*).

Mathematical models have been successfully used in studying the dynamics of infectious disease outbreaks (*Hellewell et al., 2020*; *Balilla, 2020*; *Li et al., 2020*; *Kucharski et al., 2020*; *Mukandavire et al., 2011*; *Collins & Duffy, 2018*; *Collins & Govinder, 2014*; *Tian et al., 2020*; *Mukandavire, Smith & Morris Jr, 2013*; *Tuite et al., 2011*). For instance, *Chen et al. (2020)* develop a mathematical model for calculating the potential transmission of COVID-19 from the original infection source to human infections. In this study, we will explore the control of COVID-19 using a mathematical model with China as a case study and the results could be useful for other countries with a COVID-19 outbreak.

## METHODS

### Model development

Some of the recommended control measures to reduce the spread of COVID-19 include: sensitization of the public on how to reduce transmission, isolation/quarantine of infectious individuals, prompt treatment of infectious individuals etc. Here, we formulate a mathematical model that takes these control measures into consideration. From the mode of transmission there is some distance at which the virus cannot be transmitted. Consider a location (country, community, etc.) and let $N(t)$ be the total population of individuals in that location whose probability of contacting the disease is greater than zero. Next, we partition $N(t)$ into five classes namely: Susceptible class ($S(t)$), Exposed class ($E(t)$), Infectious class ($I(t)$), Quarantine class ($Q(t)$) and Recovered/removed/death class

**Table 1** Variables and their meaning for model (1).

| Variables | Meaning |
|-----------|---------|
| $S(t)$ | Susceptible individuals at time $(t)$ |
| $E(t)$ | Exposed individuals at time $(t)$ |
| $I(t)$ | Infectious individuals at time $(t)$ |
| $Q(t)$ | quarantine/Isolated individuals at time $(t)$ |
| $R(t)$ | Recovered/Removed/Death individuals at time $(t)$ |
| $N(t)$ | Individuals in the location whose probability of becoming infected is greater than zero |
| $W(t)$ | Pathogens in the environment |

$(R(t))$. Individuals enter the susceptible class through migration/movement or birth at a rate $\Lambda$. Susceptible individuals become exposed by direct person-to-person contact at a rate $\beta_I$. Susceptible individuals can also be exposed through contact with virus in the environment (shed by persons or animals) at a rate $\beta_W$. These exposed individuals move to the infectious class at a rate $\sigma$. Infectious individual recover at a rate $\gamma$ and die from the disease at a rate $\delta$. There is also a natural death rate $\mu$. $W(t)$ represent the density of pathogens in the environment that cause the disease. Infectious individuals shed pathogen into the environment at a rate $\nu$ and these decay at a rate $\xi$.

Several control measures are introduced. Sensitization on methods that reduce an individuals chance of being infected (regular hand washing etc.) is assumed to reduce the number of susceptible individuals at a rate $\phi$. Lock-down reduces the contact rates between the susceptible individuals and infectious individuals at a rate $\varepsilon$. Individuals found to be infected are quarantined at a rate $\rho$. Based on these assumptions, the mathematical model that describes a COVID-19 outbreak is given by

## Model development

$$\frac{dS}{dt} = \Lambda - (1-\varepsilon)b_I I(t)S(t) - b_W W(t)S(t) - (\mu + \phi)S(t),$$

$$\frac{dE}{dt} = (1-\varepsilon)b_I I(t)S(t) + b_W W(t)S(t) - (\mu + \sigma)E(t),$$

$$\frac{dI}{dt} = \sigma E(t) - (\mu + \rho + \delta + \gamma)I(t),$$

$$\frac{dQ}{dt} = \rho I(t) - (\mu + \delta + \gamma)Q(t),$$

$$\frac{dR}{dt} = \phi S(t) + (\delta + \gamma)(I(t) + Q(t)) - \mu R(t),$$

$$\frac{dW}{dt} = \nu I(t) - \xi W(t). \tag{1}$$

The meaning of variables and parameters are given in Tables 1 and 2.

Since humans and pathogens have different space and time scales, it was necessary to non-dimensionalized model (1). To non-dimensionalize the model, we set the following: $\Lambda = \mu N, s = \frac{S}{N}, e = \frac{E}{N}, i = \frac{I}{N}, q = \frac{Q}{N}, w = \frac{\xi W}{\nu N}, \beta = b_I N, \alpha = \frac{b_W \nu N}{\xi}$. Based on these, the

**Table 2  Parameters and their meaning for model (1).**

| Parameters | Meaning |
| --- | --- |
| $\Lambda$ | Recruitment rate into $S(t)$ |
| $b_I$ | Contact rate with $I(t)$ |
| $b_W$ | Contact rate with $W(t)$ |
| $\mu$ | Natural death rate of humans |
| $\delta$ | COVID-19 induced death rate |
| $\rho$ | Rate of quarantine of $I(t)$ |
| $\xi$ | Pathogens decay rate |
| $\nu$ | Shedding rate of pathogens |
| $\sigma$ | Rate at which $E(t)$ becomes $I(t)$ |
| $\gamma$ | Rate of recovery for $Q(t)$ and $I(t)$ |
| $\phi$ | Rate of reduction of $S(t)$ due to sensitization information |
| $\varepsilon$ | Rate of reduction in contact rate of $S(t)$ with $I(t)$ due to lock down |

dimensionless model is given by

$$
\begin{aligned}
\frac{ds}{dt} &= \mu - (1-\varepsilon)\beta i(t)s(t) - \alpha w(t)s(t) - (\mu+\phi)s(t), \\
\frac{de}{dt} &= (1-\varepsilon)\beta i(t)s(t) + \alpha w(t)s(t) - (\mu+\sigma)e(t), \\
\frac{di}{dt} &= \sigma e(t) - (\mu+\rho+\delta+\gamma)i(t), \\
\frac{dq}{dt} &= \rho i(t) - (\mu+\delta+\gamma)q(t), \\
\frac{dr}{dt} &= \phi s(t) + (\delta+\gamma)(i(t)+q(t)) - \mu r(t), \\
\frac{dw}{dt} &= \xi(i(t) - w(t)).
\end{aligned}
\tag{2}
$$

## Model analyses

The model is used to investigate the dynamics of a COVID-19 outbreak. First, the basic reproduction number, denoted by $\mathcal{R}_0$, is calculated as epidemiologically it represents the expected number of secondary infections that result from introducing a single infectious individual into an otherwise susceptible population (*Van den Driessche & Watmough, 2002*). If $\mathcal{R}_0 < 1$, the outbreak is likely to eventually end. On the other hand, if $\mathcal{R}_0 > 1$ the outbreak will continue. Thus, the value of the basic reproduction number gives an indication on the overall rate at which the disease spreads.

The basic reproduction number ($\mathcal{R}_0$) of model (2) computed using the next generation matrix approach (*Van den Driessche & Watmough, 2002*) is

$$
\mathcal{R}_0 = \frac{\mu\sigma(\alpha+(1-\varepsilon)\beta)}{(\mu+\phi)(\mu+\sigma)(\mu+\rho+\delta+\gamma)}.
\tag{3}
$$

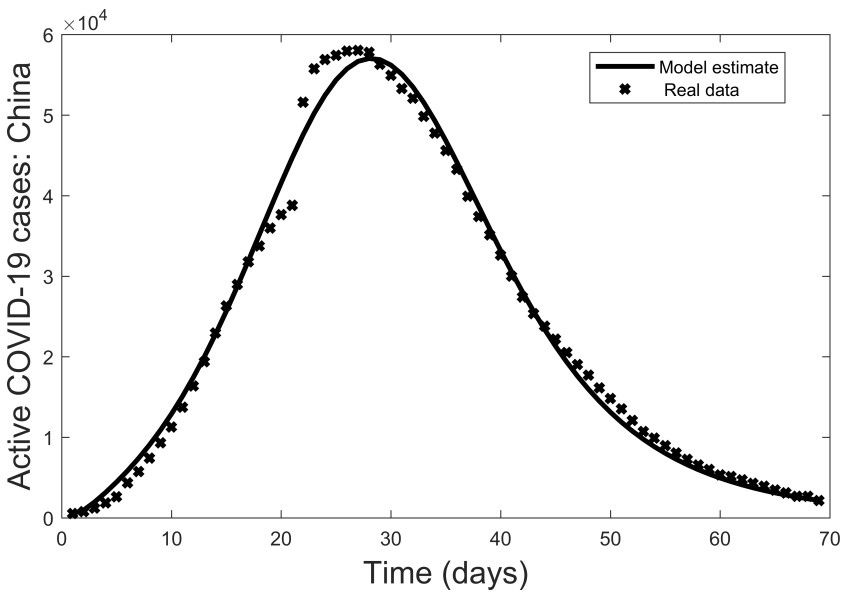

**Figure 2** Model fit of active COVID-19 cases in China from 22 January 2020 to 30 March 2020, where bold lines represent the model fit and stars mark the active cases (*Worldometer, 2020*).

The equilibrium point of model (2) when there is no disease in the population (disease free equilibrium (DFE)) is given by

$$(s^0, e^0, i^0, q^0, r^0, w^0) = \left(\frac{\mu}{\mu+\phi}, 0, 0, 0, 0, \frac{\phi}{\mu+\phi}, 0\right). \tag{4}$$

Second, the model is fit to the real data (active cases of COVID-19 *Worldometer (2020)*). In fitting the model, the parameters $\gamma, \mu, \delta, \xi$ were fixed with values taken from *Chen et al. (2020)* except $\delta$ which we estimated based on a high estimate of the death rate due to Covid-19 (i.e., Italy at the peak of their epidemic). The values are: $\gamma = 0.172$, $\mu = 0.002$, $\delta = 0.113$, $\xi = 0.100$. To estimate the remaining parameters the data is also non-dimensionalized to allow for a best fit to the model. The resulting parameter values are: $\beta = 1.097$, $\alpha = 0.098$, $\phi = 0.006$, $\sigma = 0.947$, $\rho = 0.242$, $\varepsilon = 0.211$. The results are then converted back to the original dimensions (Fig. 2). In this way the actual data is a benchmark for the ensuring visual comparisons. In terms of the overall trend, there is a reasonable fit of the data for active cases of the COVID-19 outbreak in China from 22 January 2020 to 30 March 2020. Thus, the model can be used further to explore the dynamics of COVID-19 outbreaks.

## RESULTS

### Basic reproduction number

Using the parameter values, the basic reproduction number for the COVID-19 outbreak in China as at 30 March 2020 is $\mathcal{R}_0 = 0.454 < 1$. If we exclude the control measures the basic reproduction number becomes $\mathcal{R}_0 = 4.155 > 1$. Also, using the model to estimate

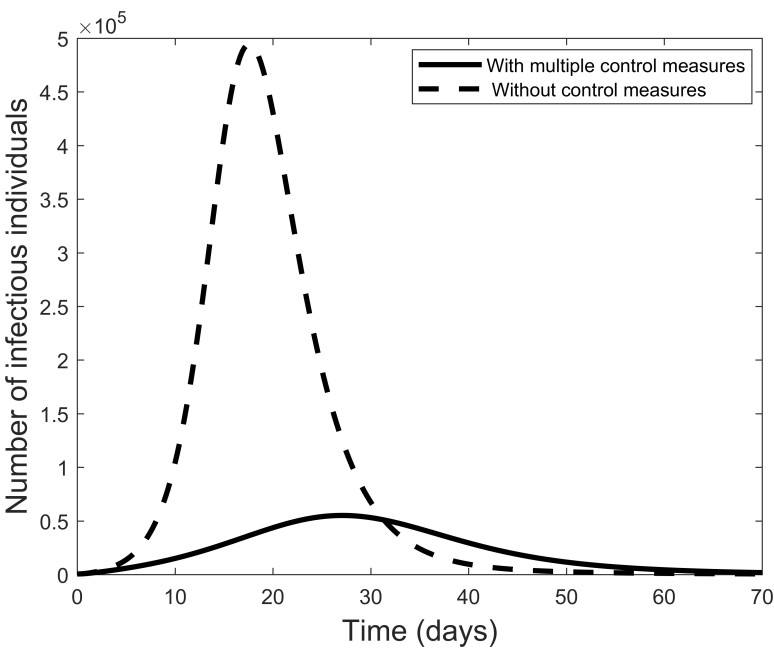

**Figure 3** Graphic illustrating the impact of using all effective control measures in fighting the COVID-19 outbreak.

the basic reproduction number for the COVID-19 outbreak in China before 20 February 2020 gives a basic reproduction number $\mathcal{R}_0 > 1$.

## Numerical simulations

Numerical simulations enable us to explore the impact of each of the control measures used in the COVID-19 outbreak in China.

Figure 3 compares the results of using all controls with the case where there are no controls. With all controls this result agrees with the case of the COVID-19 outbreak in China where the outbreak is coming to an end. On the other hand, if no control measures are used, Fig. 3 shows that many more individuals could become infectious within a short period.

Next we explore the impact of each of the various control measures. Figure 4 is a comparison of the impact of each separate control as compared to implementing all or no controls. From the figure, each control on its own reduces the number of infectious individuals but not to the extent of using all controls. Quarantine is the best individual control. When all controls are used the number quarantined is compared to those people removed from the system in Fig. 5. From the figure, the quarantining effort is greatest in the early part of the outbreak and is associated with a steady rise in individuals no longer susceptible.

The effect of implementing any two of the controls is presented in Fig. 6. Still quarantine is the essential control and together with lock-down the result more closely matches that of all controls. However, the case of sensitization together with lock-down is also significantly
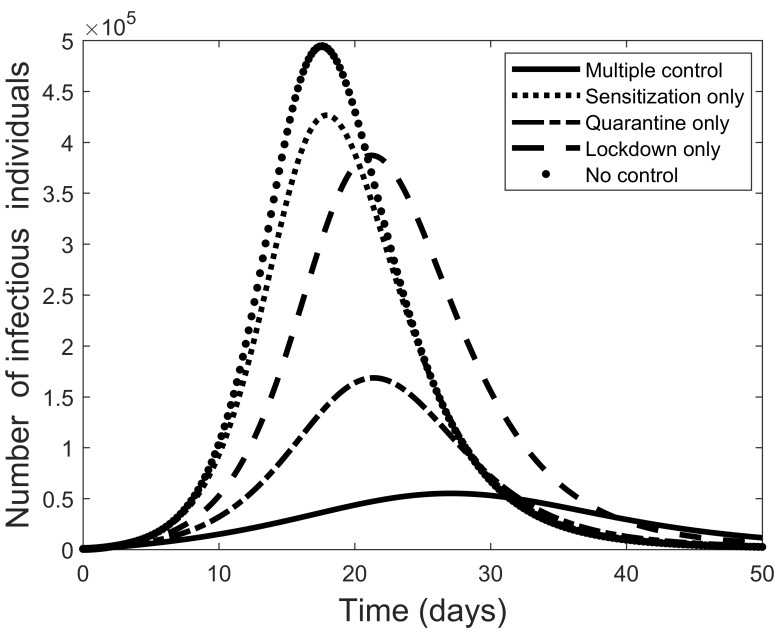

**Figure 4** Comparison of using each separate control, and the case where all control measures are used, to fight the COVID-19 outbreak.

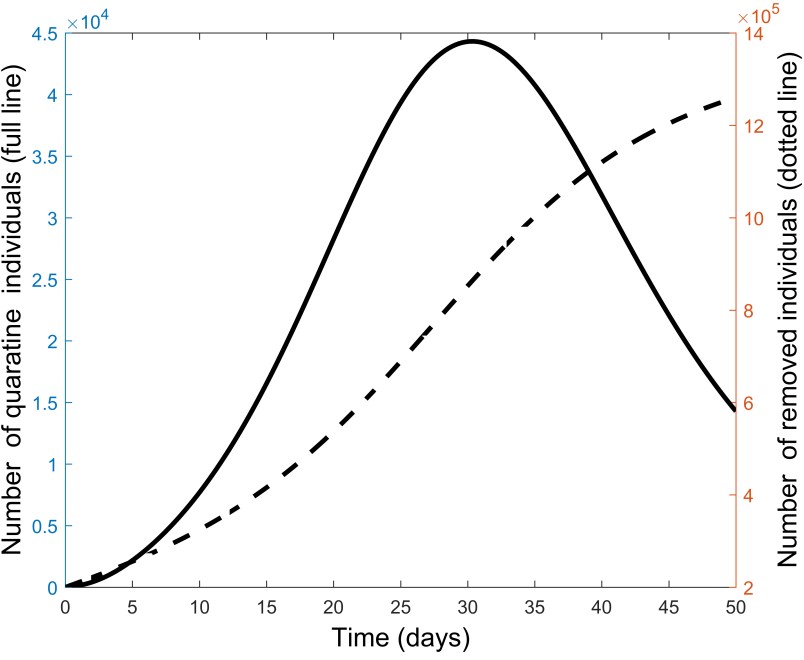

**Figure 5** Comparison of quarantine and recovered individuals over time when all control measures are combined to fight the COVID-19 outbreak.

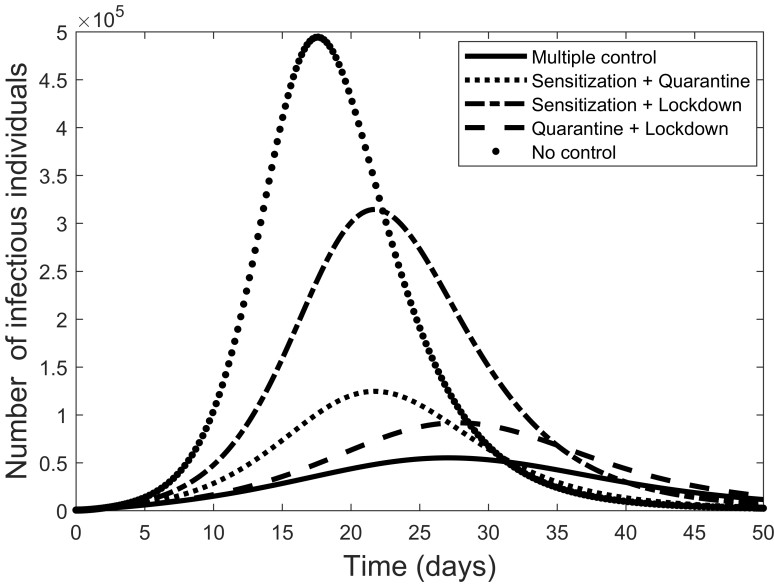

**Figure 6** Comparison of combining different control measures, and the case where all control measures are used, to fight the COVID-19 outbreak.

more effective. Further simulations, not presented here, showed that increasing any one of the separate controls can improve their effectiveness.

## DISCUSSION

COVID-19 is one of the worst infectious outbreaks currently affecting humanity globally. The disease started in China around December 2019 and has spread to almost every country of the world within a very short period. The emotional and economic impact of this outbreak could be significant. Hence the necessity to urgently stop this outbreak.

Several control measures have been recommended by the World Health Organization and other policy makers. Considering existing data on the COVID-19 outbreak in China has demonstrated that it is possible to reduce the spread of this disease and how they went about this control can be useful for other countries currently facing COVID- 19 outbreaks. In this study, data from China was used to develop a mathematical model for a COVID-19 outbreak that includes the primary control measures used.

First, epidemiological features of the model were examined using the basic reproduction number. The control parameters of the model were estimated by fitting the model to the COVID-19 outbreak data from China (from 22 January 2020 to 30 March 2020). Using these fitted parameter values, the basic reproduction number for the COVID-19 outbreak in China as at 30 March 2020 was $\mathcal{R}_0 < 1$, showing that China, through their various control intervention measures, could end the outbreak. Calculating $\mathcal{R}_0$ before 20 February 2020 (i.e., before the COVID-19 outbreak in China started decreasing) gave a value greater than unity as found in other studies (*Tian et al., 2020*; *Balilla, 2020*). Using all the data up to 30 March resulted in an $\mathcal{R}_0 > 1$, confirming that the control measures implemented in

China were effective in COVID-19 control and, where possible, should be implemented in other countries.

Second, by varying control parameters, numerical simulations are used to investigate levels of impact of the various control measures in reducing the spread of a COVID-19 outbreak. For instance, if all control measures are effectively used the spread of the disease reduces considerably in the population. On the other hand, if no control measures are considered, our result showed that a lot more of the population can be infectious within a short period. As is well understood, in this scenario the resulting curve will result in numerous deaths and extreme pressure on health care systems. To avoid these negative effects the aim is to flatten the curve as shown for the complete control scenario.

Next, each control was applied separately to measure their effect. In each case, the level of control was prescribed by the data fitting. Only quarantining of infectious individuals has the necessary effect of flattening the curve. Thus, while the other controls are helpful and the combined controls provide the best overall control, our study suggests that isolation of infectious individuals is by far the most important control method. This result agrees with the situation in South Korea where extensive testing allowed control of new infections and largely contained the outbreak (*Balilla, 2020*). On the other hand, it might be possible that the effective use of masks could have a similar effect in preventing transmission from infectious individuals that have not yet shown symptoms. Both methods of isolating infectious people were used in China (*Feng et al., 2020*; *Chen et al., 2020*; *WHO, 2020a*).

The results here suggest that hygiene behaviour is the least effective strategy. Although increasing each of the controls could increase their effect. Therefore, efforts are still required to educate individuals on disease prevention strategies. Forcing much of the public to keep separate by a strict lock down policy can also increase levels of control. For less developed countries, where isolation of infectious individual is difficult to apply effectively, the results here suggest that eventually lock down and personal prevention strategies applied together could lead to the end of an outbreak. However, this process would take longer adding to the associated financial burdens on individuals and the state.

All models depend on their simplifying assumptions. For example, the importance of population network structure could be important in how a disease spreads with this knowledge enabling the potential for targeted vaccine control (*Eubank et al., 2004*). However, in cases such as COVID-19 where there is yet no vaccine the global approach used here can be a guide. In particular, possible trends can be compared, rather than attempting to predict actual numbers, as was the objective here.

A study of the public health interventions of the COVID-19 outbreak in Wuhan, China found that over time multifaceted interventions improved control of the outbreak (*Pan et al., 2020*). Part of this intervention strategy was the robust quarantine and treatment policy of the Chinese government. A rapid review of 29 studies (10 modelling studies on COVID-19, 4 observational studies and 15 modelling studies of SARS and MERS) largely agrees with our results (*Nussbaumer-Streit et al., 2020*). These model results are also limited by the modelling assumptions used but these studies found quarantine to the most effective measure in reducing infected numbers and deaths due to COVID-19

(*Nussbaumer-Streit et al., 2020*). They also showed that combining quarantine with other control measures had a greater effect than quarantine alone, as found here.

## CONCLUSIONS

In China they managed to stem the spread of their COVID-19 outbreak. Using the Chinese data and assumptions on how the outbreak was controlled we show using an ordinary differential equations model the possible effects of different controls. Sensitization and lock-down are helpful in controlling the disease but those would need to be strongly enforced. It seems unlikely that efforts to increase these individual controls could be significantly improved beyond the efforts in China. Thus, this study points strongly to the importance of implementing all available controls. Also, the isolation of infectious individual is possibly of particular importance which would point to the need for extensive testing due to the possible prevalence of asymptomatic COVID-19 cases.

## ACKNOWLEDGEMENTS

We acknowledge and regret the pain and suffering that this virus has caused.

### Funding
The authors received no external funding for this work.

### Competing Interests
The authors declare there are no competing interests.

### Author Contributions
- Obiora C. Collins and Kevin J. Duffy conceived and designed the methods, performed the model analyses, analyzed the data, prepared figures and/or tables, authored or reviewed drafts of the paper, and approved the final draft.

### Data Availability
The raw data came from the Worldometers site; both the raw data and Matlab code are available in the Supplemental Files.

### Supplemental Information
Supplemental information for this article can be found online at http://dx.doi.org/10.7717/peerj.9933#supplemental-information.

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
