# Peer review of "Estimating the impact of lock-down, quarantine and sensitization in a COVID-19 outbreak: lessons from the COVID-19 outbreak in China"

_PeerJ, doi:10.7717/peerj.9933_

## Round 0.1 · original submission · Minor Revisions

Thank you for raising so important problem in the publication. Sorry for some delay in reviewing - we have to balance between the reviewing quality and fast manuscript processing. So, we had waited last reviewer till now. Please consider the references suggested by Reviewer #2 and mention SARS-CoV-2 as suggested by the reviewer #4. Please note recent publications on quarantine measures published last month. It is worthy to cite it to summarize the experience after the outbreak. Waiting resubmission of the manuscript.

Reviewer 1 ·

Basic reporting

The manuscript is written clear, well structures, with good illustrations and extensive (213!) references.

Experimental design

Very good experimental support for the model and the way how manuscript is designed.

Validity of the findings

Very important findings. Due to the urgent situation on understanding the best strategies in dealing with COVID-19, I think, the message of the paper and the model should be available for community and manuscript should be published ASAP.

Additional comments

I believe this is a good and important work helping to discover the strategies in managing COVID-19 pandemics.

Reviewer 2 ·

Basic reporting

In this paper authors have developed a mathematical model on COVID-19 outbreak. They have compared the model simulation result with the real data collected from china. They claimed that the quarantine (isolation) is the most effective way to control the disease. The analysis are interesting and realistic. The paper can be accepted for publication after major revision.


I have the following suggestions:

1. Organization of paper is not good.
2. Authors should mention the section number (for example 1. Introduction, 2. Methods,
2.1 Model development etc.).
3. Authors should replace Model analyses section by Equilibrium points and Basic reproduction
number.
4. Detail calculation for the evaluation of basic reproduction number is missing.
5. Endemic equilibrium point is missing.
6. Authors should check the local and global stability of the disease free and endemic equilibrium point.
7. Authors have drawn different figures to study the effectiveness of control parameters but the optimal
control problem is not defined.
8. Author should introduce the detail calculations for optimal control problem.
9. Authors should display the data collected from China in numerical simulation section.
10. Before revision, authors are advised to study and introduce the following papers in reference section

South Asian J. Math 4 (2), 69-84 (2014)
Fuzzy Information and Engineering 9 (3), 381-401(2017)
Computational and Applied Mathematics 37 (2), 1330-1351(2018)
Biophysical Reviews and Letters 14 (01), 27-48(2019)
International Journal of Nonlinear Sciences and Numerical Simulation 19(6), 627–
642(2018)
International Journal of Bifurcation and Chaos 30 (04), 2050054 (2020)

Experimental design

no comment

Validity of the findings

The authors should display the data within the manuscript.

Additional comments

Standard of your paper will be increased if you can successfully implement the suggestions. Best of luck!

Annotated reviews are not available for download in order to protect the identity of reviewers who chose to remain anonymous.

Reviewer 3 ·

Basic reporting

The paper is well written and presented, also easy to understand even to the audience outside computational/mathematical modeling field.

Experimental design

There has been previous studies focusing on using computer-based modeling to address the onset and development of COVID-19 outbreaks in different parts of the world, including China etc.. However, It is interesting to the authors presenting modeling data relating to the effectiveness of multiple control measure regarding COVID-19. Drawing real data from China as basis of the presented modeling and providing insights to other parts of the world seems feasible and meaningful.

It seems reasonable for the authors to try to answer a medical-social problem by conduct such mathematical simulation. Especially when the measures recommended by authorities are being questioned and some of the recommendations in different countries even contradict in certain degree.

Validity of the findings

Nevertheless, it has been a common sense that the 3 key issues to prevent any infectious/contagious disease should always be: 1) Controlling the source of infection; 2) Cutting off the channels of transmission; and 3) Protecting susceptible/vulnerable population. The last one could be reached by immunization or vaccination, which in current case, would take around one year to develop a vaccine. Therefore, I would suggest the authors categorize different control measures taken and examine whether the approach examined is effective (could further elaborate in the Discussion Part).

Given the nature that SARS-CoV-2 is highly contagious, it would be less likely to control the source of infection, and cutting off the channels of transmission (e.g. quarantine and isolation of confirmed COVID-19 individuals, etc.) seemed more effective, which is consistent with what the manuscript has presented so far. I would believe that by linking these medical interpretations with the mathematical contents would make the article for powerful and convincing.

Additional comments

Please add more infectious disease science or public health science related content in the Discussion Part to make the article a stronger one.

Reviewer 4 ·

Basic reporting

I have pointed out in the attached annotated document where improvements in the language may be done

Experimental design

The authors need to review their use of the term susceptibility in the document. My argument is that the model captures this word as reducing the number of susceptible individuals and that is why they leave the susceptible class direct to the removed class due to sensitization. However the word is always used in the sense of likelihood of an individual being infected.Therefore if they claim sensitization reduces susceptibility then it means individuals can still be infected but at with a reduced likelihood and should therefore not be taken directly to the removed class since this suggests they can never be infected at all.

Validity of the findings

No comment

Additional comments

I suggest you make reference to the actual name of the virus SARS-CoV-2 that causes COVID 19 at least once in the document

Annotated reviews are not available for download in order to protect the identity of reviewers who chose to remain anonymous.

---

## Round 0.2 · accepted · Accept

Thank you for the update. There is some minor remark about susceptibility rate (reviewer #2). See Line 61 and Table 3. I believe you'd correct it without additional review. However, there are no more critical comments. I recommend accept it now.

Reviewer 3 ·

Basic reporting

no comment

Experimental design

no comment

Validity of the findings

In the revised manuscript, the authors have added a paragraph comparing their work to other public health science related content, which I believe is a good addition.

Reviewer 4 ·

Basic reporting

.

Experimental design

.

Validity of the findings

.

Additional comments

I believe the manuscript is ok though the authors have not addressed my concern on susceptibility (see line 61).

Annotated reviews are not available for download in order to protect the identity of reviewers who chose to remain anonymous.